# Play with Me: How Fathers and Mothers Play with Their Preschoolers with Autism

**DOI:** 10.3390/brainsci13010120

**Published:** 2023-01-10

**Authors:** Silvia Perzolli, Arianna Bentenuto, Giulio Bertamini, Paola Venuti

**Affiliations:** 1Laboratory of Observation, Diagnosis, and Education (ODFLab), Department of Psychology and Cognitive Science, University of Trento, 38068 Rovereto, Italy; 2Department of Child and Adolescent Psychiatry, Pitiè-Salpêtrière Hospital, Sorbonne University, 75651 Paris, France

**Keywords:** autism spectrum disorder (ASD), play, mother–child interaction, father–child interaction

## Abstract

(1) Background: Children can develop cognitive and social skills during play. Most research has focused on mothers, but the paternal features in interaction with children with autism spectrum disorder (ASD) are mainly unexplored. This study aimed to compare fathers’ and mothers’ interactive behaviors with their children with ASD to identify similarities and differences during playful exchanges. (2) Methods: A total of 72 mothers and 72 fathers of paired children with ASD (chronological age: M = 44.61 months; SD = 13.37) took part in this study. Data were collected during 10 min of video-recorded semi-structured interactions with mothers and fathers separately in interaction with their children. (3) Results: Mothers showed more symbolic play (W = 3537; *p* < 0.001) than fathers, who displayed higher levels of exploratory play (t(139.44) = −2.52; *p* = 0.013) compared to mothers. However, child cognitive functioning impacts maternal play but not the father’s play characteristics. (4) Conclusions: Highlighting mother–child and father–child features may have important service delivery implications for implementing personalized parental-based interventions based on the strengths and weaknesses of both caregivers in a complementary system.

## 1. Introduction

### 1.1. Play in the Context of Autism Spectrum Disorder

The development of play behaviors permits the exploration of the child’s world, finding information about the child’s cognitive development, and exploring the child’s affective and social domains [1]. Given the social nature of play, the observation of play behaviors is also fundamental to detect aspects of caregiver–child interaction.

Since play evolution is related to cognitive abilities and language [2], impairments in the child’s cognitive functioning may lead to less sophisticated play. However, more advanced levels in child’s play are achieved during the interplay with an adult both in children with typical and atypical development [2,3,4,5,6,7].

Although play allows social engagement with others, both play and social behaviors are particularly impaired in children with autism spectrum disorder (ASD) [2,8,9]. ASD is a neurodevelopmental condition in which children show impairments in the socio-communicative area with a restricted and repetitive pattern of behaviors [10]. ASD symptomatology also affects children’s play, which appears repetitive with poor creative engagement with objects [9]. In addition, children with ASD show impairments in the symbolical use of objects, representing one of the crucial clinical aspects observed during the assessment of preschool children [11]. In addition, the play behaviors of children with ASD tend to focus more on the atypical use of objects as a source of sensory stimulation rather than for interactive and social intentions [12]. Some findings report that children with ASD spend less time in functional activities than children with typical development and Down syndrome [13]. The altered play behaviors’ trajectory may also be influenced by the lack of joint attention skills [14,15], lack of imitation skills [16], and the presence of sensory hypo-hyper sensitivity activation [17]. In fact, in children with ASD, there might be an increased or decreased reactivity to sensory inputs or atypical interests in sensory elements in the environment. These alterations can preclude knowledge of a specific play’s function and the ability to interact with another person during a shared play activity.

### 1.2. Mother–Child and Father–Child Interaction in the Context of ASD

Early caregiver–child interactions are essential for the child’s typical and atypical development [18,19,20]. However, socio-communicative deficits of children with ASD may alter the creation of good relationships in the caregiver–child dyad, causing maladaptive interactive exchanges that need to be appropriately re-adapted through targeted developmental interventions. Children with ASD show a tendency to be less responsive and more focused on objects than on partners [8,21]. These children tend to display fewer involving attempts towards others, rejecting or ignoring their parents’ social initiatives, more often than children with typical development [22]. The literature highlights the need to support parents during exchanges with children with ASD and, therefore, the need to involve both parents effectively in intervention programs with their children.

Mothers of children with ASD tend to use more intrusive behaviors (e.g., calling the child’s name frequently; proposing different activities without considering the child’s intentionality; or interrupting the child’s play) than mothers of children with typical development. Furthermore, mothers also seem to show fewer social and verbal attempts directed at their preschool children with ASD [23]. A recent study investigated the play behaviors of mothers with children diagnosed with ASD, mothers with children diagnosed with Down syndrome (DS), and mothers of typically developing children and revealed that mothers with children diagnosed with ASD followed their child’s lead during activities equally to mothers in the two other conditions. In addition to this, other research pointed out that mothers of children with ASD seemed to show fewer symbolic activities directed at their children. Despite this, a positive association between the amount of maternal symbolic attempts and children’s symbolic play was found only in children with ASD, shedding light on the critical role of mothers in the child’s cognitive development, especially in dyads with young children with ASD [24].

Despite the well-known impact of healthy father–child relationships on the child’s development [25], emotional regulation [26], and symbolic play [6], research on parenting in the context of ASD has mostly focused on mothers rather than fathers. Furthermore, strategies that fathers used to support the child’s activities reduced the child’s externalizing problems [27].

Studies conducted so far showed that fathers of children with ASD tended to look at their children more often than fathers of children with typical development [6]. They also engaged more frequently in physical contact activities with the child than fathers of children with Down syndrome and typical development. Further, fathers of children with ASD addressed more play proposals to the child compared to fathers of children with TD. A comparison among fathers of children in different conditions revealed that fathers of children with ASD had greater suggestions and intrusive behaviors directed to the child than fathers of children with DS and TD [28].

### 1.3. Comparison between Mothers and Fathers of Children with ASD

Some research examined the comparison between fathers and mothers of children with ASD, revealing that both parents display comparable levels of engagement when interacting with their children [29]. However, other studies reported differences in caregivers’ behaviors when interacting with their children with ASD [30]. Fathers of children with ASD seemed to exhibit more intrusive behaviors than fathers of children in other atypical conditions and typical development [29], even if the intrusive style does not emerge when paternal behaviors are compared to maternal ones. In this case, mothers displayed more intrusive and controlling behaviors than fathers [31]. Moreover, fathers displayed greater social and physical engagement, while mothers used more supportive strategies (e.g., language) to interact with the child [32]. Both caregivers showed similar levels of engagement with their children [29]. Furthermore, both mothers and fathers tend to display the same frequency of simple and complex strategies to support their children’s regulation. The children, in turn, reached out to the two caregivers for help with comparable frequency [33,34]. In line with this, some findings described similar levels of affective behaviors exhibited by the two parents [35] and the same number of responsive behaviors and strategies to start activities with their children. However, despite some similarities, other studies showed differences in parents’ behaviors during interactions with their children with ASD [30]. Some research reported lower attention and initiation shown by fathers of children with ASD in comparison with mothers [30,36]. In addition, other studies revealed that fathers seemed to be less supportive, showing lower levels of involvement compared to mothers during the interaction [32], pointing out the need for more empirical research. With respect to the child’s behaviors in interaction with the caregivers, different results emerged in the literature. In fact, some literature pointed out that children seem to engage the two caregivers for a similar quantity of time during the interaction [35,37]. However, other studies reported that children seemed to show different levels of involvement between mothers and fathers [25].

To conclude, given the impact of parents on child development, it is critical to systematically investigate the interaction modalities in order to support caregivers in personalized, parental-based interventions and improve impaired interactive exchanges. Parental-based interventions aim to support parental behaviors during interaction with their children while supporting targeted children’s behaviors (e.g., social and communicative impairments). Considering the fundamental role of parents’ involvement during the therapeutic intervention with their preschoolers [38], it is important to examine similarities and differences in parenting behaviors when interacting with their children.

### 1.4. The Current Study

#### Aims and Hypotheses

The characteristics of play of children with ASD has received some empirical attention in recent years; however, play characteristics of fathers and mothers when interacting with their children were less explored, and there are some contrasting findings in the literature that need additional investigation.

With this in mind, this study aimed to investigate father–child play during interaction compared to mother–child play. Further, we aimed to explore how child characteristics of cognitive functioning and symptom severity impact the implementation of play activities by fathers and mothers. Specifically, we aimed to do the following:Explore the play behaviors of each caregiver separately, expecting mothers to display higher levels of symbolic play than exploratory play. We hypothesized that fathers show an opposite trend of behaviors with a preference for exploratory play activities than symbolic ones;Compare father–child and mother–child play behaviors when interacting with their children, expecting that fathers display lower levels of symbolic play compared to mothers, in line with previous research depicting mothers as more focused on implementing supportive strategies [32]. On the contrary, we expected fathers to show greater levels of exploratory play than mothers when interacting with their children, given their primary interest in physical and social play;Examine the child’s behaviors considering differences in play concerning the two caregivers. We expect children to show more symbolic play with mothers, consistent with the idea that mothers may show higher levels of symbolic play [24,31]. However, considering the internal structure of child’s play, we hypothesized that they would show higher levels of exploratory play than symbolic play with both caregivers given the specific impact of ASD core symptoms on symbolic activities and their preference for exploratory activities;Investigate the ability of fathers and mothers to adapt their play to the play displayed by their children. To do so, we explored associations between fathers’ and mothers’ playful behaviors to play levels of the child. In particular, we expected that both caregivers’ behaviors could adapt to child characteristics of play showing similar levels of symbolic and exploratory play;Examine the impact of both child cognitive functioning and child symptom severity on levels of play exhibited by the two caregivers. Specifically, given the strong interconnection between play and cognitive abilities, we expected that the degree of cognitive impairment might impact the implementation of play activities and strategies in both caregivers.

## 2. Materials and Methods

### 2.1. Participants and Procedure

A total of *n* = 144 participants participated in this study, including 72 mother–child dyads and 72 father–child dyads. Overall, 72 children (67 males and 5 females) with ASD (chronological age: M = 44.61 months; SD = 13.37) in interaction with their fathers (paternal age M = 39.39 years; SD = 6.09) were compared to the same children in interaction with their mothers (maternal age M = 38.39 years; SD = 5.34). Children in this study were in the preschool age range (2–6 years old). All data were collected at ODFLab (Laboratory of Observation, Diagnosis, and Intervention) of the Department of Psychology and Cognitive Science, University of Trento (Italy). ODFLab is a research and clinical center where families or adults turn to have a functional diagnosis for themselves or their children and/or undertake a therapeutic program. Children’s diagnosis of ASD was confirmed through clinical observation by a trained psychologist on the basis of the DSM-5 criteria for autism spectrum disorder and through the administration of the Autism Diagnostic Observation Schedule (ADOS-2, [39] see measures), the golden-standard instrument for the diagnosis. Participants were recruited voluntarily through advertisements in the waiting room of the ODFLab. Later, a research referent not involved in the clinical evaluation scheduled a meeting with the families to explain the research objectives and the overall procedure. Families who agreed to participate in the study signed a written consent form. The research was conducted in compliance with the last version of the Declaration of Helsinki [40]. The Ethics Committee of the University of Trento (Italy) approved all the procedures (protocol number 2020-042).

Play behaviors were assessed through 10 min of video-recorded interactions with fathers and mothers separately using a standard set of toys (train; toy car; toy phone; dinette set made of glasses, cutlery, mocha, mugs, saucers, and pans; doll; ball; puzzle box; books). The toy set is made of objects familiar to all children in their domestic context and was considered according to various child developmental levels and allows a wide variety of different play routines. Parents were asked to play spontaneously with their children as they would do in their domestic context for ten minutes. Interactions took place during different sessions at the beginning of the clinical evaluation in order to avoid possible child’s distress during the clinical evaluation. In general, fathers and mothers played with their children in different sessions with at least 7 days in between based on parents’ availability, mitigating the possible effect of the order. In the other cases (when parents came the same day) the parents’ order was randomized.

In addition, interactions were coded through a standardized observative tool that assesses dyadic play behaviors (the Play Code, [41]). Two independent observers codified the videos of mother–child and father–child interactions after attending a training for the application of the Play Code and reaching a significant level of interrater reliability.

The videos were randomly distributed to the observers. The average kappa between coders for the play levels ranged from 0.79 to 0.88 for training videos.

The interactions took place in the context of clinical assessment and were conducted during different sessions at the beginning of the clinical evaluation. Clinical assessments were made by several meetings of clinical observations and administration of standardized tests to define the clinical profile and diagnosis.

The trained observers used BORIS (Behavioral Observation Research Interactive Software) [42], an open-source event logging software for video/audio coding and live observations. This software allows continuous coding of play levels as well as start and end times (accurate to 1 ms). The child’s level of symptom severity was assessed through the ADOS-2 [39]. Further, the Griffiths Mental Development Scale—Edition Revised (GMDS-ER, [43] see measures) was used to assess children’s general developmental quotient.

### 2.2. Measures

#### Play Code

To assess dyadic play, we considered “the Play Code”, which considers the sophistication level of play abilities [3,41,44]. The Play Code is made by a mutually exclusive and exhaustive category system characterized by different play levels and a default category of non-play. The different play levels came from empirical research on the developing nature of play during the first years of life. Each play level is described in the table below (see Table 1). Thanks to the Play Code, it is possible to determine which play activity is performed by the individual more frequently and can be used as an index of play sophistication. The Play Code allows to determine whether or not symbolic activities are present during the exchange and which specific level of symbolization the child shows.

Furthermore, the Play Code evaluates the play activity’s duration, a critical element to infer the attention and concentration level and to have an overall idea of the child’s play abilities. The Play Code is applied to both parents’ and child’s play, revealing several aspects since it permits a simultaneous but at the same time independent coding of caregiver and child play. For this, it allows us to examine if and how different play levels are associated with different activities. See Table 1 for descriptions of play levels.

The time threshold was set at 1 s. Then, a play level was coded until a ten seconds’ break was identified. During the breaks, none of the partners had to touch an object. Four metrics were computed for each play level: absolute and proportional frequencies and absolute and proportional durations. The scores were then normalized through the proportion of maximum scaling (“POMS”), a min-max normalization method [45,46] that ranges from 0 to 1. The formula was: POMS(x) = [(x − min)/(max − min)]. This method produces a monotonic metric limiting the problems related to standardization in longitudinal design [45]. A general play index for each level was computed by averaging the POMS (i.e., absolute and proportional frequencies and absolute and proportional durations). Afterward, the POMS indexes were grouped into two global indexes: exploratory play, i.e., the average of levels from 1 to 4, and symbolic play, i.e., the average of levels from 5 to 8. The indexes represent quantitative measures of the frequency and duration of the different types of play, accounting for the absolute and proportional measures of the interplay.

### 2.3. Autism Diagnostic Observation Schedule-2 (ADOS-2)

ADOS-2 [39] is used to assess the child’s symptoms severity during the first functional and diagnostic assessment. This instrument is a golden-standard instrument for the diagnosis of ASD and is designed to be used with individuals referred for possible autism spectrum disorders. This instrument allows a diagnostic classification and outlines a behavioral pattern of the child. Further, this tool is internationally used and validated [39,47]. For this study, we used mainly Toddler Module, Module 1, and Module 2 for children with no language or language made by single words and simple phrases. The scores are organized into two macro-categories: social affect (SA), which includes communication and social interaction features, and repetitive and restrictive behaviors (RRB). Each module gives a total score for the ADOS diagnostic classification (autism–autism spectrum–non-spectrum). This score is transformed into the comparison score—used as the severity score [48]—that classifies the severity of symptoms into three categories (mild, moderate, or severe). Trained psychologists carried out the administration of this tool after an official ADOS course.

### 2.4. Griffiths Mental Development Scales

The Griffiths Mental Development Scales—Edition Revised (GMDS-ER, [43] are developmental scales used to assess children from birth to 8 years old through semi-structured activities designed to evaluate different aspects of children’s mental development. GMDS-ER provides an estimate of the child’s developmental age (DA) and developmental age in different domains resulting in a general developmental quotient (GQ). The scales provide Z-scores from 1 to 5 for children ranging from 0 to 24 months, and the sixth scale is for children older than two years, specifically (1) locomotion; (2) personal and social; (3) language and communication; (4) hand–eye coordination; (5) performance; and (6) practical reasoning. All scores are standardized (M = 100; SD = 15).

### 2.5. Analytic Plan

Statistical analyses were performed using *R* Software [49]. First, descriptive statistics of data were calculated, reporting means and standard deviations of each level of play of fathers, mothers, and their children, as well as considering the indexes of exploratory and symbolic play. Second, data were checked for normality and linearity through the *Shapiro-Wilk and Levene’s tests*. Comparisons between the two caregivers were performed using independent samples *t*-tests or Wilcoxon–Mann–Whitney tests in case of non-parametric statistics of the data. Paired *t*-tests or Wilcoxon signed-rank tests were applied to investigate differences in child’s play considering fathers and mothers. Associations between paternal and maternal play to child’s play were calculated using the Pearson correlation coefficient. Correlations were also applied to examine associations between the child’s development and severity characteristics with the child’s play and the play of both caregivers, respectively.

## 3. Results

### 3.1. Comparison between Parents

The *t*-test for independent samples or Wilcoxon–Mann–Whitney tests revealed a significant difference in the levels of symbolic play (W = 3537; *p* < 0.001) between the two caregivers. In line with our hypotheses, mothers displayed higher levels of symbolic play (M = 0.13; SD = 0.06) compared to fathers (M = 0.077; SD = 0.72). When considering the index of exploratory play, our results show a significant difference between the two caregivers (t(139.44) = −2.52; *p* = 0.013), with fathers (M = 0.15; SD = 0.07) demonstrating higher levels compared to mothers (M = 0.12; SD = 0.06). These differences referring to whole indexes are also confirmed by the duration of the exchange. In fact, the duration of symbolic play in fathers (M = 40.51; SD = 45.54) is significantly (W = 3499; *p* < 0.001) lower than the duration of symbolic play displayed by mothers (M = 79.57; SD = 72.67). On the contrary, the duration of paternal exploratory play is higher (M = 123.89; SD = 91.10) than the duration of exploratory play of mothers (M = 86.56; SD = 70.56) (W = 1921.5; *p* < 0.007).

Paired-sample *t*-tests or paired Wilcoxon–Mann–Whitney tests also revealed significant differences in child symbolic play exhibited with the two caregivers (V = 1061; *p* = 0.006), with higher levels exhibited when playing with mothers (M = 0.08; SD = 0.09) compared to fathers (M = 0.05; SD = 0.07). No differences emerged in the child’s levels of exploratory play with the two caregivers. Interestingly, when analyzing the internal play structure of each caregiver, children showed higher levels of exploratory play both with fathers (t(71) = 7.84; *p* < 0.001) and with mothers (t(71) = 4.74; *p* < 0.001) compared to symbolic play. In line with this, we analyzed the symbolic and exploratory play of each caregiver, revealing that, in line with children, fathers display higher levels of exploratory play compared to symbolic play (t(71) = 5.53; *p* < 0.001), whereas mothers show similar levels of symbolic and exploratory play. See Table 2.

### 3.2. Association between Parent and Child Play

Moreover, our results revealed an association between the child index of exploratory play with the exploratory play of fathers (r = 0.57; *p* = 0.01) and the exploratory play of mothers (r = 0.49; *p* = 0.01). In addition, children’s symbolic play is associated with both the symbolic play of mothers (r = 0.52; *p* = 0.01) and fathers (r = 0.33; *p* = 0.01). Results revealed a low correlation between the duration of child exploratory play and mothers’ duration of exploratory play (r = 0.26; *p* = 0.05), but a moderate correlation was found concerning the duration of exploratory play of fathers (r = 0.42; *p* = 0.01). Further, the total duration of a child’s play, which includes all the activities and levels of play, is associated with the total amount of time of fathers’ play (r = 0.34; *p* = 0.01), but it is not associated with the total duration of the mother’s playtime.

### 3.3. Association between Child Characteristics and Play

To conclude, we wanted to investigate the relationship between the child’s characteristics of cognitive functioning and symptom severity and the aspects of play considering the child as well as fathers and mothers. With respect to the child, higher levels of cognitive functioning (considering each subscale of cognitive development expect to motor-gross area) are associated with higher levels of symbolic activities (r = 0.48; *p* = 0.01) but not with exploratory play when playing with both mothers and fathers.

Interestingly, child levels of exploratory play are negatively associated with language and communication subdomain (r = −0.32: *p* = 0.008) and performance subscale (r = 0.32; *p* = 0.05) when playing with mothers but not with fathers.

Concerning mothers’ play, children’s general cognitive functioning is not associated with mothers’ levels of exploratory play. However, they may tend to exhibit higher levels of symbolic play with children with higher levels of general cognitive functioning (r = 0.26; *p* = 0.05), especially in the subscale of language and communication (r = 0.33; *p* = 0.007) and hand–eye coordination (r = 0.43; *p* = 0.005).

Interestingly, no associations were found between children’s cognitive functioning and the father’s exploratory or symbolic play level. When considering the exchange duration, children with higher cognitive levels display symbolic play for extended periods both with mothers (r = 0.37; *p* = 0.01) and fathers (r = 0.42; *p* = 0.01).

Concerning the severity of symptoms, we found that children with more severe symptomatology showed less symbolic play with fathers (r = −0.40; *p* = 0.05) but not with mothers. However, the symptomatology did not seem to impact the levels of play displayed by the two caregivers in their interaction with the children. The total duration of the child’s play is influenced by the degree of symptoms of ASD both when interacting with mothers (r = −0.43; *p* = 0.05) and fathers (r = −0.52; *p* = 0.01), revealing that when symptoms are higher, children tend to play for less time with both caregivers. In addition, for the duration, no differences emerged concerning caregivers’ play according to the severity of child’s symptoms.

## 4. Discussion

Given the scarcity of studies comparing paternal and maternal play behaviors with children with ASD and the urge to examine these aspects, with this work, we wanted to provide an empirical effort to delineate the play behaviors of fathers and mothers. Generalizing previous research results is difficult due to the lack of a large sample size of fathers and mothers of children with ASD. Further, the paucity of highly standardized observational tools may prevent a clear and coherent profile of parental characteristics, undermining consistent and robust results [50].

In line with our hypothesis, mothers compared to fathers displayed higher levels of symbolic play. These findings are consistent with previous literature underlying their didactic and teaching role when interacting with their children, which previously emerged both in the context of typical and atypical development [51,52,53,54,55]. Mothers prefer more structured and complex activities that enhance the child’s cognitive aspects [6]. On the other hand, fathers seem to favor more simple, social, and less object-oriented activities that aim to increase the child’s social skills [54,56,57]. In our findings, fathers displayed higher levels of exploratory play rather than symbolic and elaborated activities. In line with our expectations, children in this sample were more likely to play in an exploratory and functional way. For this, fathers seemed to follow the children’s guide rather than propose sophisticated and object-mediated play.

Paternal adaptation to the child’s behaviors is confirmed by the fact that fathers exhibited higher levels of exploratory play than symbolic play coherently with their children’s proposals. In contrast, mothers showed similar internal levels of symbolic and exploratory play. Some peculiar differences emerge when the child’s play is compared between fathers and mothers. In fact, given the higher amount of symbolic activity exhibited by mothers, children displayed a more significant amount of symbolic play when interacting with them compared to their fathers. This is also in line with previous findings showing that greater skills in symbolic play are reported when play is scaffolded and supported by an adult [58]. However, when both child indexes of play are compared (exploratory and symbolic), it emerged that children seem to prefer exploratory activities rather than symbolic ones with both fathers and mothers, in line with the impact that core symptoms of ASD have on symbolic play. As expected, children tend to engage more in exploratory activities, which appears to be their main channel to interact with a playmate.

Further, when the duration of child and parent play is considered, the child’s length of play is only associated with the father’s duration but not with the maternal one. In this sense, fathers seem to follow more what the child proposes without enhancing the levels of play but letting themselves be guided by the child’s intentions. Interestingly, it emerges that the two caregivers fulfill different but equally important functions for the child’s development, strengthening the complementary role of both parents in the development of different functions. On the one hand, the major levels of symbolic stimulation displayed by mothers may impact the child’s level of cognitive activities. On the other hand, fathers play a fundamental role in developing the child’s intentionality. In addition, maternal symbolic play but not paternal symbolic play seems to be influenced by the child’s cognitive impairments. However, the child’s symptoms severity does not impact the play modalities of both parents. In line with previous findings, mothers are more focused on teaching cognitive skills, and therefore, they may be more affected by their child’s cognitive deficits [59].

Despite the importance of investigating both parents’ interactive features in the context of ASD, some limitations should be considered. First, our sample is unbalanced regarding child gender, preventing the opportunity to differentiate play modalities of boys and girls and parents with their daughters and sons. Second, we did not compare parents of children with typical development or other clinical conditions. Third, participants in our study may be vulnerable to reactivity; in fact, participants may modify some aspects of their behaviors when they know they are video-recorded.

Future research should also include parents in different conditions to better consider other possible intervening factors in the comparison that may be specific to ASD. Third, this work has a cross-sectional design; therefore, longitudinal studies are needed to effectively investigate the impact of parental behaviors on child characteristics of cognitive functioning and symptom severity. Finally, it would be interesting to investigate the possible presence of paternal and maternal autistic traits in the context of parent–child interaction and assess their impact on child variables.

## 5. Conclusions

Despite some limitations, this work may have relevant implications in providing clinicians and practitioners with more information about fathers’ and mothers’ specific behaviors when interacting with their children. Exploring the specific features of both fathers and mothers in interaction with their children may help to better tailor the activities during interventions based on the interactive style of both caregivers. Fathers seemed to display a relationship-oriented approach, characterized by simple play more focused on social engagement and on the child’s intentionality. On the other hand, mothers showed behavioral traits that go in the direction of the child’s cognitive development, addressing more information while speaking and more sophisticated play while playing. These differences highlighted the presence of two distinct patterns of behavior that characterize parents in the context of ASD even if intervention programs are usually tailored to maternal traits. An optimized intervention should, therefore, implement different activities considering the two caregivers, working on both paternal and maternal weaknesses but starting from their strengths. Furthermore, given the fundamental role of fathers in social engagement with their children, understanding and considering the paternal interactive behaviors when playing with their children may have important service delivery implications for effectively involving them in play-based early interventions for children with ASD.

## Figures and Tables

**Table 1 brainsci-13-00120-t001:** Descriptions of play levels.

Play Level	Definition	Example
Level 1 Unitary functional activity	Refers to a single functional action focused on the production of effects that were unique to a single object	E.g., kicking the ball
Level 2 Inappropriate combinatorial activity	Refers to incorrect combinatorial action, including the inappropriate juxtaposition of two or more objects	E.g., putting the fork on the train
Level 3 Appropriate combinatorial activity	Refers to proper combinatorial activity, including the correct association of two or more objects	E.g., putting the spoon on the saucer
Level 4 Transitional play	Reflects the first approach of pretend play	E.g., pretend to eat without any vocalizations
Level 5 Self-directed pretense	Considers pretense play activity directed toward self	E.g., pretend to eat from an empty plate with vocalizations
Level 6 Other-directed pretense	Considers pretense play activity directed toward others	E.g., feeding the doll
Level 7 Sequential pretense	Refers to the ability of linking two or more pretense actions	E.g., pouring into an empty cup from the teapot and then drinking
Level 8 Substitution pretense	Refers to the substitution of objects with others	E.g., using the fork as a brush
Level 9 Non-Play	Refers to non-play activities	The child or the adult neither play with nor touch an object; e.g., the child is running afinalistically around the room

**Table 2 brainsci-13-00120-t002:** Descriptive and Inferential Statistics of Paternal and Maternal Play.

Play	Fathers		Mothers		
	Mean	SD	Mean	SD	*p*-Value
Child Exploratory Play Index	0.16	0.63	0.15	0.58	t(71) = −0.58; *p* = ns
Child Symbolic Play Index	0.05	0.07	0.08	0.09	V = 1061; *p* = 0.006 **
Parent Exploratory Play Index	0.15	0.07	0.12	0.06	t(139.44) = −2.52; *p* = 0.013 *
Parent Symbolic Play Index	0.08	0.72	0.13	0.06	W = 3537; *p* < 0.001 ***
Total Child Play Index	0.10	0.38	0.12	0.42	t(71) = 2.94; *p* = 0.004
Total Parent Play Index	0.11	0.42	0.13	0.45	t(141.441) = 1.68; *p* = ns
Total Duration Child Play	283.43	165.44	296.64	138.45	t(71) = 0.77; *p* = ns
Total Duration Parent Play	164.40	104.56	166.13	102.41	W = 2599.5; *p* = ns
Duration Child Exploratory Play	236.64	147.67	223.82	135.29	t(71) = −0.74; *p* = ns
Duration Parent Exploratory Play	123.89	91.09	86.56	70.56	W = 1921.5; *p* = 0. 007 **
Duration Child Symbolic Play	46.79	66.59	72.82	90.59	V = 995; *p* = 0.014
Duration Parent Symbolic Play	40.51	45.54	79.57	72.67	W = 3499; *p* < 0.001 ***

** p* < 0.05; ** *p* < 0.01; *** *p* < 0.001.

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
