# Peer review of "Play with Me: How Fathers and Mothers Play with Their Preschoolers with Autism"

_brainsci, 2023, doi:10.3390/brainsci13010120_

Round 1

Reviewer 1 Report

The article contains a good discucssion of the symptoms in terms of difficulties in children with ASD in he form of play activities. A shallow diagnosis of ASD, without evalutaion of cognitive functioning, and without differential diagnosis for other conditions that produce symptoms idential to ASD, poses a high risk of diagnostic error. A good critical view of the Authors for the study conducted. Methodologically correct research process. Interesting in the cognitive aspect of a hitherto enigmatically explored topic.

Author Response

<<Dear Reviewer, thank you so much for your comment, we really appreciated your thoughts about the manuscript>>

Reviewer 2 Report

Thank you for the opportunity to review the manuscript titled “Play with Me: How Fathers and Mothers Play with their Preschoolers with Autism”. This study assessed the play behaviors of parents of children with ASD and the relationship between the parent play behaviors and the child play behaviors and child cognition and symptom severity. I enjoyed reading this manuscript and found the study both interesting and of value to researchers, service providers, and parents alike. As I have outlined in my comments and questions below, there are a number of issues that must be addressed before this manuscript might be considered for publication. I have included both minor considerations (often concerns with wording and simple errors that can be revised easily) and major considerations (i.e., items that may preclude publication) for each section of the manuscript. I hope that these are helpful.

Introduction:

Minor considerations:

·         Pg 1, Line 33. Explain what you mean by “play progress”. This will help the reader understand what you are referring to.

·         Pg 1, Line 35. Explain what you mean by “favored” or choose a different term.

·         Pg 1, Line 41. Reword the sentence beginning with “Further…”. It is a little awkward.

·         Pg 2, beginning at Line 57. Avoid the terms “circuits” and “re-activated” unless you are referring to neurophysiology. If you are, then this should be fully explained.

·         Pg 2, beginning at Line 62. This is not a sentence. Reword.

·         Pg 2, Line 66. Explain, describe, or provide examples of “intrusive behaviors”.

·         Pg 2, Lines 70-71. Explain what adapting to a child’s activities means.

·         Pg 2, beginning at Line 77. The sentence beginning with “As for children…” does not make sense. Revise or clarify.

·         Pg 2, Lines 80-81. Explain what “essential consequences” means.

·         Pg 2, Lines 93-95. This sentence is awkward and lacks clarify. Please revise.

·         Pg 3, Lines 108-112. This is a run-on sentence. It is also vague and needs further clarification.

·         Pg 3, Line 119. This is a conditional statement with a hypothetical (i.e., If this research was conducted, then…), but the research has not been hypothetical. Eliminate this sentence or reword to only indicate that less research has been conducted on play characteristics of parents and what has been done includes contrasting findings.

·         Pg 3, Line 139. Replace “coherently” with “consistent”.

Major considerations:

·         The introduction includes good background research, but it does not flow well. Consider revising it to begin with mother-child play behaviors (including both well-documented and inconsistent findings), then discuss father-child play behaviors (including both well-documented and inconsistent findings), before summarizing how mothers and fathers interact differently in play and the need for further research.

·         Something that needs to be conveyed in the introduction is why differences in parent play engagement matters. I think the implications of finding any differences needs to be more explicit. There is some reference to this in the conclusion, but indicating the value of this research should be included in the introduction as well.

Methods:

Minor considerations:

·         Pg 1, Line 154. Clarify that the 144 dyads includes 72 mother-child dyads and 72 father-child dyads.

·         Pg 5, Lines 201-218. I suggest creating a table for this information.

·         Pg 5, Line 232. Should “amount” be “frequency”?

·         Pg 5, Line 241. Change “module Toddler” to “Toddler Module” and capitalize Module each time.

·         Pg 6, Line 254. I think “estimate” should be substituted for “exact”. There is no indication that any measure of cognition is “exact”.

Major considerations:

·         The sample (i.e., 10 minutes) is very short and could be vulnerable to reactivity. I’m concerned that there is no mention of the brevity of the observation (is there precedence?) and no attempt to reduce reactivity. At the very least, this should be noted in the limitations within the discussion. Additionally, while I understand the importance of standardization, using the same toys across all children may restrict their play behaviors. It would be beneficial to describe the types of toys that were available.

·         More details about how the play session was conducted would be helpful. For example, what instructions were given (were they standardized?).

Results:

Minor considerations:

·         Pg 6, Line 280. Substitute “demonstrating” for “reporting” as there were no “reports” of play.

·         Pg 6, Line 290. Please clarify that this finding (i.e., higher levels of exploratory play with both parents) was a difference between exploratory and symbolic play.  

Major considerations:

·         The statistics in the table do not match in-text statistics (e.g., Child Symbolic Play Index [p=.47 or p=.0004?]; Parent Exploratory Play Index [p=.13 or p=.013?]; Duration of Parent Exploratory Play [p=007 or p=.007?]). Additionally, based on some of the in-text statistics, the SDs are massive in relationship to the means. This calls into question the accuracy of the statistical analyses.

Discussion:

Minor considerations:

·         Pg 8, Line 337. I assume that this should be “parental” and not “paternal” as both parents’ characteristics are important.

·         It appears that the limitation of small sample size is noted in the first paragraph, but that detracts from the findings. I suggest moving any mention of sample size to the end of the discussion.

·         Pg 8, Lines 341-342. Is the statement regarding a mother’s preference for structured and complex activities based upon this study? If so, that would be inaccurate, as this study did not measure parent preference. If it is based on previous research, please cite that research.

·         Pg 8, Lines 358 and 360. Like the previous comment, can you say that children “prefer” exploratory activities? If some of the child’s play is influenced by parent play behaviors then I don’t think you can report that the child prefers a particular type of play.

·         Pg 8, beginning with Line 362. I’m not sure that this is accurate. Please clarify. Just by looking at the total duration of child play for each parent (283 +/- 165 and 296 +/- 138, father vs. mother, respectively) and total duration for parent play (164 +/- 104 and 166 +/- 102), I don’t see how there would be a relationship between child and father, but not child and mother.

·         Pg 8, beginning at line 370. It seems plausible, if not likely that child cognitive level is a mediating variable as children with lower cognitive abilities may be less likely to engage in symbolic play, which in turns influences mothers’ choice to engage in symbolic play.

·         Pg 9, Lines 382-384. It’s unclear what is meant by “child characteristics” and “effectively”. Please clarify.

Major considerations:

·         None

Author Response

#Reviewer 2

Thank you for the opportunity to review the manuscript titled “Play with Me: How Fathers and Mothers Play with their Preschoolers with Autism”. This study assessed the play behaviors of parents of children with ASD and the relationship between the parent play behaviors and the child play behaviors and child cognition and symptom severity. I enjoyed reading this manuscript and found the study both interesting and of value to researchers, service providers, and parents alike. As I have outlined in my comments and questions below, there are a number of issues that must be addressed before this manuscript might be considered for publication. I have included both minor considerations (often concerns with wording and simple errors that can be revised easily) and major considerations (i.e., items that may preclude publication) for each section of the manuscript. I hope that these are helpful. 

<< Dear Reviewer, 

Thank you for the opportunity to revise our manuscript. We are grateful for all the comments and suggestions provided for this work. We believe that the work benefited from all the comments enhancing the power and readability. 

We addressed all the reviewers' comments and provided a new version of the manuscript with tracked changes. >>

Introduction:

Minor considerations:

  •         Pg 1, Line 33. Explain what you mean by “play progress”. This will help the reader understand what you are referring to.

<<Thanks for the comment. We changed play progress into play evolution to provide more understanding>>

  •         Pg 1, Line 35. Explain what you mean by “favored” or choose a different term.

<<Thanks for the comment. We rephrase the sentence as follows: However, more advanced levels in child’s play are achieved during the interplay with an adult both in children with typical and atypical development>>

  •         Pg 1, Line 41. Reword the sentence beginning with “Further…”. It is a little awkward.

<<Thanks for the comment. We rephrased as follows: ASD symptomatology also affects children’s play that appears more repetitive with poor creative engagement>>

  •         Pg 2, beginning at Line 57. Avoid the terms “circuits” and “re-activated” unless you are referring to neurophysiology. If you are, then this should be fully explained.

<<Thanks. We changed circuits into interactive exchanges and re-activated into restored>>

  •         Pg 2, beginning at Line 62. This is not a sentence. Reword.

<< Thanks for the comment. We rephrased as follows: These children tend to display fewer involving attempts towards others, rejecting or ignoring their parents’ social initiatives more often than children with typical development>>

  •         Pg 2, Line 66. Explain, describe, or provide examples of “intrusive behaviors”.

<<Thanks for pointing this out. We provided examples in brackets al follows (e.g., calling often the child’s name; propose different activities without considering the child’s timing; interrupting the child’s play) >>

  •         Pg 2, Lines 70-71. Explain what adapting to a child’s activities means.

<<Thanks for the opportunity to explain this better. We rephrased the sentence as follows: ]. A recent study comparing mothers of children with ASD with mothers of children with typical development and Down Syndrome (DS) revealed that mothers of children with ASD followed their child’s lead during activities equally to mothers in the two other samples. >>

  •         Pg 2, beginning at Line 77. The sentence beginning with “As for children…” does not make sense. Revise or clarify.

<<Thanks for the comment. We rephrased the sentence as follows: Despite the well-known importance of healthy father-child relationships on the child’s development, emotional regulation [25, 26], and symbolic play [6], research on parenting in the context of ASD focused on mothers rather than fathers>>

  •         Pg 2, Lines 80-81. Explain what “essential consequences” means.

<< Thanks, we changed “essential consequences” into had an impact>>

  •         Pg 2, Lines 93-95. This sentence is awkward and lacks clarify. Please revise.

<<Thanks, we tried to provide more clarity. Even if fathers seem to display more intrusive behaviors compared to fathers of children in other atypical conditions and typical development [29], the intrusive style does not emerge when paternal behaviors are compared to maternal ones.>>

  •         Pg 3, Lines 108-112. This is a run-on sentence. It is also vague and needs further clarification.

<< Thanks. We rephrased as follows: the different results in the literature are confirmed by the child’s behaviors in interaction with caregivers. In fact, some literature revealed that children  seem to engage parents for the same amount of time during the exchange [35, 37], but other works pointed out that children seemed to show different levels of involvement between mothers and fathers [25] >>

  •         Pg 3, Line 119. This is a conditional statement with a hypothetical (i.e., If this research was conducted, then…), but the research has not been hypothetical. Eliminate this sentence or reword to only indicate that less research has been conducted on play characteristics of parents and what has been done includes contrasting findings.

<< Thanks for pointing this out. We rephrased as suggested. Characteristics of play of children with ASD received some empirical attention in recent years, however, play characteristics of fathers and mothers when interacting with their children were less explored, and there are some contrasting findings in the literature that need additional investigation>>

  •         Pg 3, Line 139. Replace “coherently” with “consistent”.

<< Thanks for the suggestion >>

Major considerations:

  •         The introduction includes good background research, but it does not flow well. Consider revising it to begin with mother-child play behaviors (including both well-documented and inconsistent findings), then discuss father-child play behaviors (including both well-documented and inconsistent findings), before summarizing how mothers and fathers interact differently in play and the need for further research.

<<Thank you for the comment. We restructured this part following your indications in presenting the mother's and father’s play styles and their comparison while evidencing well-documented and non consistent findings that need more research. See the revised manuscript.>>

  •         Something that needs to be conveyed in the introduction is why differences in parent play engagement matters. I think the implications of finding any differences needs to be more explicit. There is some reference to this in the conclusion, but indicating the value of this research should be included in the introduction as well.

<< Thanks for the suggestion. We highlighted this aspect at the end of the introduction. See main manuscript >>

Methods:

Minor considerations:

  •         Pg 1, Line 154. Clarify that the 144 dyads includes 72 mother-child dyads and 72 father-child dyads.

<<Thank you for the comment, we rephrased as follows to provide more clarity: ​​N=144 participants took place in this study, including 72 mother-child dyads and 72 father-child dyads.>>

  •         Pg 5, Lines 201-218. I suggest creating a table for this information.

<< Thanks for the suggestion, we provided the table considering different play levels

Table 1. Descriptions of play levels 

Play Level

Definition

Example

Level 1

Unitary functional activity

refers to a single functional action focused on the production of effects that were unique to a single object

e.g., kicking the ball

Level 2

Inappropriate combinatorial activity

refers to incorrect combinatorial action, including the inappropriate juxtaposition of two or more objects

e.g., putting the fork on the train

Level 3

Appropriate combinatorial activity

refers to proper combinatorial activity including the correct association of two or more objects

e.g., putting the spoon on the saucer

Level 4

Transitional play

reflects the first  approach of pretend play

e.g., pretend to eat without any vocalizations

Level 5

Self-directed pretense

considers pretense play activity directed toward self

e.g., pretend to eat from an empty plate with vocalizations

Level 6

Other-directed pretense

considers pretense play activity directed toward others

e.g.,feeding the doll

Level 7

Sequential pretense

refers to the ability of linking two or more pretense actions

e.g., pouring into an empty cup from the teapot and then drinking

Level 8

Substitution pretense

refers to the substitution of objects with others

e.g., using the fork as a brush

Level 9 

Non-Play

refers to non-play activities

the child or the adult don’t play nor touch an object. E.g., the child is running afinalistically around the room

  •         Pg 5, Line 232. Should “amount” be “frequency”?

<<Thanks for noticing this, we modified this is the text>>

  •         Pg 5, Line 241. Change “module Toddler” to “Toddler Module” and capitalize Module each time.

<<Thanks, we modified this in the main text>>

  •         Pg 6, Line 254. I think “estimate” should be substituted for “exact”. There is no indication that any measure of cognition is “exact”.

<<Thanks for the suggestion, we modified this in the manuscript>>

Major considerations:

  •         The sample (i.e., 10 minutes) is very short and could be vulnerable to reactivity. I’m concerned that there is no mention of the brevity of the observation (is there precedence?) and no attempt to reduce reactivity. At the very least, this should be noted in the limitations within the discussion. Additionally, while I understand the importance of standardization, using the same toys across all children may restrict their play behaviors. It would be beneficial to describe the types of toys that were available.

<<Thank you for the opportunity to describe this more in detail. First of all, considering the duration of the interactions we added specific references in favor of the fact that ten minutes may be sufficient to display spontaneous interactive and playful behaviors. Several studies indicate that using ten-min observation constitute a valid temporal parameter (see Bornstein et al. 1996). Furthermore, we used ten minutes interactions also in consistency with other observational research in the field ( Ostfeld-Etzion et al., 2016).

Furthermore, we added the reactivity in the limitations section as follows: Third, participants in our study may be vulnerable to reactivity. 

Furthermore, thanks for the opportunity to explain better the toys provided to the participants. We added the following sentences in the manuscript: 

Play behaviors were assessed through ten minutes of video-recorded interactions with fathers and mothers separately using a standard set of toys (train, toy car, toy phone, dinette set made by glasses, cutlery, mocha, mugs, saucers and pans, doll, ball, puzzle box and books). The toy set is made of objects familiar to all children in their domestic context and was considered according to various child developmental levels and allows a wide variety of different play routines >>

  •         More details about how the play session was conducted would be helpful. For example, what instructions were given (were they standardized?)

<< Thanks for noticing this, additional information are provided in the main manuscript as follows: parents were asked to play spontaneously with their children as they would do in their domestic context for ten minutes>>

Results:

Minor considerations:

  •         Pg 6, Line 280. Substitute “demonstrating” for “reporting” as there were no “reports” of play.

<<Thanks for the suggestion. We modified this in the main manuscript>>

  •         Pg 6, Line 290. Please clarify that this finding (i.e., higher levels of exploratory play with both parents) was a difference between exploratory and symbolic play.

<<Thanks for noticing this. We modified this in the main manuscript>>

Major considerations:

  •         The statistics in the table do not match in-text statistics (e.g., Child Symbolic Play Index [p=.47 or p=.0004?]; Parent Exploratory Play Index [p=.13 or p=.013?]; Duration of Parent Exploratory Play [p=007 or p=.007?]). 

Additionally, based on some of the in-text statistics, the SDs are massive in relationship to the means. This calls into question the accuracy of the statistical analyses.

<<This is a very important comment and thanks to this we had the opportunity to provide more accurate results.

As you correctly pointed out, play indices show high standard deviations. We further inspected the data to detect the presence of possible outliers. Observing the data, the variability is mainly related to differences in play profiles, especially in the symbolic dimension, with many children showing no symbolic play opposed to a subset of children with good symbolic ability. As pointed out by correlation analysis, these abilities are also associated with parents’ play profiles. Given the developmental age of the sample, this is coherent with the clinical picture of autism with reference to play development. To account for this, in our data analysis we checked data for normality through the Shapiro test. In case of non-normality of data, we performed the nonparametric Wilcoxon-Mann-Whitney test, that makes no assumptions on the initial distribution of data. However, while re-checking the data to address the comment of the reviewer, we noticed a coding error in the R script for normality check. Therefore, some of our variables violated the normality assumption and remained undetected in the first version. We thank the reviewer for the possibility of discovering this error and we updated the manuscript and the analysis accordingly. In summary, some variables (and in particular, the variables with high variances underlined by the comment) were now tested with the non-parametric test. The results remained coherent with the ones that emerged before but were verified with non-parametric inferential tests more robust to the observed data distributions.  >>

Discussion:

Minor considerations:

  •         Pg 8, Line 337. I assume that this should be “parental” and not “paternal” as both parents’ characteristics are important.

<<Thanks for the comment >>

  •         It appears that the limitation of small sample size is noted in the first paragraph, but that detracts from the findings. I suggest moving any mention of sample size to the end of the discussion.

<<Thanks for noticing this. We rephrase the sentences at the beginning of the discussion to provide more clarity.>>

  •         Pg 8, Lines 341-342. Is the statement regarding a mother’s preference for structured and complex activities based upon this study? If so, that would be inaccurate, as this study did not measure parent preference. If it is based on previous research, please cite that research.

<<Thanks for the comment. We provided a reference considering this>>

  •         Pg 8, Lines 358 and 360. Like the previous comment, can you say that children “prefer” exploratory activities? If some of the child’s play is influenced by parent play behaviors then I don’t think you can report that the child prefers a particular type of play.

<<Thanks for the comment. We modified the text to improve clarity>> 

  •         Pg 8, beginning with Line 362. I’m not sure that this is accurate. Please clarify. Just by looking at the total duration of child play for each parent (283 +/- 165 and 296 +/- 138, father vs. mother, respectively) and total duration for parent play (164 +/- 104 and 166 +/- 102), I don’t see how there would be a relationship between child and father, but not child and mother

     <<Thanks for the comment. We checked again the data and it emerged a correlation between father total duration of play and child’s play duration but not with mothers, despite the single durations being comparable with respect to means and standard deviations. We better precised these in the discussion, highlighting that our consideration is grounded in the correlation analysis. We also report the Pearson’s correlation coefficients with its p-value for the correlation between the mother’s total play duration and the child’s one (r=0.16; p=0.16).>> 

  •         Pg 8, beginning at line 370. It seems plausible, if not likely that child cognitive level is a mediating variable as children with lower cognitive abilities may be less likely to engage in symbolic play, which in turns influences mothers’ choice to engage in symbolic play.

      <<Thanks for the comment. We definitely agree with this, however, it was interesting for us to note that the correlation was only present with mothers’ symbolic play but not with fathers, probably confirming their different play behaviors (being mothers more focused on symbolic activities, they may be more influenced by cognitive levels of children compared to fathers)>>

  •         Pg 9, Lines 382-384. It’s unclear what is meant by “child characteristics” and “effectively”. Please clarify.

<< Thank you for noticing this, we rephrased as follows: Third, this work has a cross-sectional design; therefore, longitudinal studies are needed to effectively investigate the impact of parental behaviors on child characteristics of cognitive functioning and symptoms severity. >>

Major considerations:

  •         None

Reviewer 3 Report

Study Title:  Play with Me: How Fathers and Mothers Play with their Preschoolers with Autism

Study Findings

The current study compared the play behaviors during the mother-child and father-child interactions. Both parents and children showed more symbolic play during mother-child than father-child interaction. On the contrary, they showed more exploratory play during father-child than mother-child interaction. Moreover, children’s characteristics are more associated with the father’s than the mother’s play behaviors. The current study highlighted the differences in play behaviors when interacting with fathers and mothers. Clinicians should consider the strength and weaknesses of both caregivers when designing parent-oriented interventions.

Abstract

1.       In Lines 18-19, the authors wrote “Mothers showed more symbolic play (t(142)=3.85; p<0.001), whereas fathers displayed higher levels of exploratory play (t(142)=-2.52;p=0.013). Could the author be more specific about what is being compared (i.e., mothers vs fathers, or symbolic vs exploratory play)?

Introduction

1.       Please make sure there is a hypothesis associated with each study aim. I think the hypothesis is missing for aim 4.

2.       For hypothesis 3, could the author specify how children’s exploratory play might differ when interacting with mothers or fathers?

Materials and Methods & Results

1.       Did the author randomize the order for mother-child or father-child interactions? If not, could the author explain how the order might affect the interactions in the discussion section?

2.       Could the author be more specific about what variables are considered parametric or non-parametric?

3.       There are many T-tests or Wilcoxon-Mann-Whitney tests conducted in this study. Did the author use any methods to control for the multiple comparisons?

4.       For the parametric comparisons, did the author consider using Dyad (father-child vs mother-child) x play type (Symbolic vs exploratory) 2-way ANOVA to explore how parents’ and children’s behaviors might be affected by the dyad and play type? I suggest first running the ANOVAs and only conducting the post-hoc t-test when a significant interaction/main effect is found.

5.       For the associations of the non-parametric variables, did the author consider using Spearman instead of the Pearson correlation?

6.       For the associations between children’s development and play behaviors, did the author correlate the GMDS-ER total score with the play behaviors? Since the GMDS-ER include not only the cognitive functions (it also includes locomotor and coordination subscales), it would be interesting to see if the standard score of each subtest correlates with the play behaviors.

Discussion

1.       Since many children with ASD have family histories of the disorder, it would be interesting to consider fathers’ and mothers’ autistic traits when investigating the parent-child interaction. Could the author comment on that and maybe add this as a future research direction?

Minor edits

1.       In Table 1, please explain what * indicates.

2.       I suggest moving 3.1 Analytic Play to the method section.

Author Response

#Reviewer 3

<<Dear Reviewer, 

Thank you for the opportunity to revise our manuscript. We are grateful for all the comments and suggestions provided for this work. We believe that the work benefited from all the comments enhancing the power and readability. 

We addressed all the reviewers' comments and provided a new version of the manuscript with tracked changes.>>

Study Findings

The current study compared the play behaviors during the mother-child and father-child interactions. Both parents and children showed more symbolic play during mother-child than father-child interaction. On the contrary, they showed more exploratory play during father-child than mother-child interaction. Moreover, children’s characteristics are more associated with the father’s than the mother’s play behaviors. The current study highlighted the differences in play behaviors when interacting with fathers and mothers. Clinicians should consider the strength and weaknesses of both caregivers when designing parent-oriented interventions.

Abstract

  1.       In Lines 18-19, the authors wrote “Mothers showed more symbolic play (t(142)=3.85; p<0.001), whereas fathers displayed higher levels of exploratory play (t(142)=-2.52;p=0.013). Could the author be more specific about what is being compared (i.e., mothers vs fathers, or symbolic vs exploratory play)?

<<Thank you for the comment. We rephrased the sentence as follow to provide more clarity in the abstract: Mothers showed more symbolic play (t(142)=3.85; p<0.001) than fathers, who displayed higher levels of exploratory play (t(142)=-2.52;p=0.013) compared to mothers.>>

Introduction

  1.       Please make sure there is a hypothesis associated with each study aim. I think the hypothesis is missing for aim 4.

<<Thanks for noticing this, we added the hypothesis in the main manuscript >>

  1.       For hypothesis 3, could the author specify how children’s exploratory play might differ when interacting with mothers or fathers?

<<Thanks for the comment. We modified the text to provide more clarity. We specify that our hypothesis was considering differences in the two parents, however, we hypothesized differences also considering the child, mainly with respect to symbolic play, given that literature supports a specific impairment in this area for children with ASD >>

 Materials and Methods & Results

  1.       Did the author randomize the order for mother-child or father-child interactions? If not, could the author explain how the order might affect the interactions in the discussion section?

<<Thank you for the comment. We clarify this aspect in the main manuscript.
Interactions took place during different sessions at the beginning of the clinical evaluation in order to avoid possible child’s distress during the clinical evaluation. In general, fathers and mothers played with their children in different sessions with at least 7 days in between based on parents’ availability, mitigating the possible effect of the order. In the other cases (when parents came the same day) the parents’ order was randomized.>>

  1.       Could the author be more specific about what variables are considered parametric or non-parametric?

<< This is a very important comment and thanks to this we had the opportunity to provide more accurate results.

As you correctly pointed out, play indices show high standard deviations. Observing the data, the variability is mainly related to differences in play profiles, especially in the symbolic dimension, with many children showing no symbolic play opposed to a subset of children with good symbolic ability. As pointed out by correlation analysis, these abilities are also associated with parents’ play profiles. Given the developmental age of the sample, this is coherent with the clinical picture of autism with reference to play development. In our data analysis we checked data for normality through the Shapiro test. In case of non-normality of data, we performed the nonparametric wilcoxon-mann-whitney test that makes no assumptions on the initial distribution of data. While re-checking the data to address your comments, we noticed a coding error in the R procedure of normality check. Therefore, some of our variables violated the normality assumption. We thank the reviewer for the possibility of discovering this error and updated the manuscript and the analysis accordingly. In summary, some variables (and in particular, the variables with high variances underlined by the comments) were now tested with the non-parametric test. The results remained coherent with the ones that emerged before but were verified with non-parametric inferential tests more robust to the observed data distributions. >>

  1.       There are many T-tests or Wilcoxon-Mann-Whitney tests conducted in this study. Did the author use any methods to control for the multiple comparisons?

<<Thanks for the comment. Multiple comparisons would be an issue for repeated testing the same variable for the same population. In our case, the only juxtaposition would regard the relationship between the total and the specific play indices, i.e., exploratory and symbolic, which form the total scores. In the other cases, symbolic and exploratory play represents different and independent dimensions, as well as quality and quantity of play. The partially related and non independent variables would therefore regard the tests of differences between total indices (total, and duration) and subdimensions (symbolic and exploratory), making pairs of N = 2 partially non independent measures. Applying Bonferroni, which is the most conservative correction method, would therefore require a p-value of p=0.025 as significance threshold. Looking at the results, the inferential tests on the total indices and durations already resulted in either non-significant or p<0.025, as well as the sub indexes for both durations and total scores.>>

  1.       For the parametric comparisons, did the author consider using Dyad (father-child vs mother-child) x play type (Symbolic vs exploratory) 2-way ANOVA to explore how parents’ and children’s behaviors might be affected by the dyad and play type? I suggest first running the ANOVAs and only conducting the post-hoc t-test when a significant interaction/main effect is found.

<<Thanks for the suggestion. We did not consider using the interaction between dyad and play type since we do not have a dichotomous variable. The nature of the measures is not mutually exclusive, and during the coding time both parents and children generally show both exploratory and symbolic play. The total indices are formed including 8 different levels of play type in terms of both frequency and duration (details are provided in the methods section. At an exploratory level we tried to incorporate a factor variable based on the presence / absence of children’s symbolic play in the interactions. In this way, we could employ the two way ANOVA, as suggested by the reviewer, on the global indices which are the only one normally distributed. For the child’s indices of total play and duration, we used repeated measures ANOVA. The analysis converges with our analytic approach. Durations are not affected by the interaction between presence / absence of child’s symbolic play in the interactions, and parent (mother / father), and no main effect emerges. Further, both the child’s and the parent’s total play indices showed both the parent main effect and the interaction parent * presence / absence of child’s symbolic play. Post-hoc comparisons confirmed that the child’s total play index is significantly higher with mothers and that the parent's total index of play is not significantly different between fathers and mothers. We can share the results of this additional analysis but preferred to leave the inferential approach described in the manuscript since the majority of the variables of interest were not-normally distributed, the ANOVA may be more complex to interpret, has more strict assumptions to verify, and the numerical nature of the indices allows to use this approach only for the general ones.>>

  1.       For the associations of the non-parametric variables, did the author consider using Spearman instead of the Pearson correlation?

<< Thanks for reporting this important issue. Pearson correlation does not require the assumption of normality and is best suitable for continuous data, which is the case for both the total indices and the durations, given their numerical nature. However, pearson’s r may be less robust than spearman's correlation coefficient if outliers are present in data. As previously noted, the normality assumption was rejected mainly due to the presence of substantial inhomogeneities in children’s and parents’ play profiles. Observing data suggests that this may mainly be due to the presence of children with low overall levels of play together with children with higher and adequate, especially in the symbolic domain, abilities, and coherently with ASD. In any case, we also performed Spearman’s correlation coefficient for the non-normally distributed variables confirming the significance of results >>

  1.       For the associations between children’s development and play behaviors, did the author correlate the GMDS-ER total score with the play behaviors? Since the GMDS-ER include not only the cognitive functions (it also includes locomotor and coordination subscales), it would be interesting to see if the standard score of each subtest correlates with the play behaviors.

<<Thanks for this comment and the opportunity to deepen the analyses. As you suggested we performed correlations also considering the subscales of the GMDS and added the significant correlations in the main manuscripts >>

Discussion

  1.       Since many children with ASD have family histories of the disorder, it would be interesting to consider fathers’ and mothers’ autistic traits when investigating the parent-child interaction. Could the author comment on that and maybe add this as a future research direction?

<<Thanks for the suggestion. We added this in the future directions>>

Minor edits

  1.       In Table 1, please explain what * indicates.

<<Thanks for noticing this. We added the caption at the end of the table>>

  1.       I suggest moving 3.1 Analytic Play to the method section.

<<We moved the 3.1 analytic plan at the end of the methods section>>

Round 2

Reviewer 2 Report

I appreciate the authors’ rapid revision and the opportunity to provide a second review. In general, the authors did a nice job of making revisions and addressed almost all of my original comments, questions, and concerns. The manuscript still needs editing at various points due to the phrasing and word choices. I have several questions related to some of the statistics reporting as well. Below are some additional edits and questions to consider:

Abstract

Given that the focus of this study is parent play behavior, I suggest reporting the number of parents in the study rather than the number of children (or report both). I don’t believe that the mean age is needed for the children as it is reported in the Methods section. I’m not sure what the authors are reporting in the abstract for the statistics. What is the “W” referring to?

Introduction:

Minor considerations:

·         Pg 1, Line 34. The change from “play progress” to “play evolution” still does not offer clarity as to what this is referring to. I think this is referring to the development of play behaviors.

·         Pg 1, Line 39. What does “fundamental space” mean? This is not clear.

·         Pg 1, Lines 43-45. I’m not sure that ASD symptomology “affects children’s play”. Rather repetitive play is one feature often observed in children with ASD.

·         Pg 2, Line 64. I understand that poor interaction during play may affect the parent-child relationship (albeit I’m not sure to what degree); however, how is this relationship “restored”? I don’t think that the relationship is “restored”. (This should also be addressed on Pg 3, Line 148).

·         Pg 2, Line 66. I would use a different word than “involving” as it reads awkwardly.

·         Pg 2, Lines 69-72. This sentence does not make sense and needs reworded.

·         Pg 2, Lines 74-75. The examples of intrusive behavior are fine, but poorly worded. I would delete “often” or change it to “…calling the child’s name frequently”. “Child’s timing” is unclear.

·         Pg 2, Line 78. The tense in this sentence is present and should be changed to past tense (i.e., investigated instead of that investigates). Additionally, the rest of the sentence needs rewording. I suggest “A recent study investigated the play behaviors of mothers with children diagnosed with ASD, mothers with children diagnosed with Down syndrome, and mothers of typically-developing children.” Then describe the findings. The current sentence describing the findings should be reworded as well.

·         Pg 2, Line 83. I don’t think “endeavors” fits here.

·         Pg 2, Line 95. I think “has” or “has mostly” should be inserted between “ASD” and “focused”.

·         The tense sometimes shifts from present to past and vice-versa. Please make sure that all sentences maintain the same verb tense throughout the introduction when discussing past research.

·         Pg 2, begging at Line 148. The review of the literature is focused on parent play behaviors, but in this sentence, it jumps to therapeutic intervention. There needs to be a better explanation of what intervention means. Is the intervention on the parents’ behaviors, the children’s behaviors, or both? What does intervention look like?

·         Pg 4, Line 154. Consider changing to “The characteristics of play of children with ASD has received…”. At is currently written the tense is incorrect and the wording is slightly awkward.

·         Pg 4, Line 174. Add a comma between “mothers” and “consistent”.

Methods:

Minor considerations:

·         Pg 4, Line 191. Consider beginning the sentence with something other than N=144. You might try “A total of 144…”. Additionally, participants do not “take place” – they participate. The following sentence beginning with “72 children” needs to be reworded.

·         Pg 4, Lines 200-201. “Confirmed” is a better choice than “proved”. Additionally, delete “and judgment”.

·         Pg 5, Lines 203-204. Delete “a golden standard instrument for the diagnosis, i.e.,”.

·         Pg 5, Line 207. Delete “dedicated”.

·         The description of the play sessions is much better than before, but could use editing and additional information.

·         Pg 5. What is the clinical evaluation that was taking place? This should be explained.

·         The description of the Play Code needs additional editing.

·         The Play Code is at times capitalized and at other times it is not. Please use the appropriate form consistently.

·         Pg 7, Paragraph beginning at 290. This paragraph needs additional editing. Some of the procedures are clear and some are not.

Results:

Minor considerations:

·         It is unclear what the W statistic being used is. Is this Kendall’s W? There is also a V statistic that is in the text and table. This needs explanation.

Discussion:

Minor considerations:

·         Pg 11, Line 439. I don’t think that the term “intricate” is a good choice. Simply using “difficult” would work. Pg 12, Line 486. Please expound upon the vulnerability to reactivity.

Author Response

I appreciate the authors’ rapid revision and the opportunity to provide a second review. In general, the authors did a nice job of making revisions and addressed almost all of my original comments, questions, and concerns. The manuscript still needs editing at various points due to the phrasing and word choices. I have several questions related to some of the statistics reporting as well. Below are some additional edits and questions to consider:

<<Dear reviewer, thank you for the opportunity to re-submit a new version of our manuscript. All your comments were beneficial for our work, and we hope the manuscript is now suitable for publication to Brain Sciences >>

Abstract

Given that the focus of this study is parent play behavior, I suggest reporting the number of parents in the study rather than the number of children (or report both). I don’t believe that the mean age is needed for the children as it is reported in the Methods section. I’m not sure what the authors are reporting in the abstract for the statistics. What is the “W” referring to?

<<Thank you very much for your comment. We added the number of parents as you suggested. However, we would like to maintain the mean age of children in the abstract to better specify in the abstract the average age of the parents’ children to whom we refer.
W is referring to the non-parametric test used to compare the two samples of mothers and fathers (Mann-Wilcoxon-Whitney Test).>>

Introduction:

Minor considerations:·         Pg 1, Line 34. The change from “play progress” to “play evolution” still does not offer clarity as to what this is referring to. I think this

 is referring to the development of play behaviors.

<<Thank you for all your suggestions. We modified in the main manuscript. see tracked changes>>

  •         Pg 1, Line 39. What does “fundamental space” mean? This is not clear.

<<Thank you. We modified the sentence in the main manuscript >>

  •         Pg 1, Lines 43-45. I’m not sure that ASD symptomology “affects children’s play”. Rather repetitive play is one feature often observed in children with ASD.

<<Thank you for the comment. Several studies indicate that children with autism present impairments in the development of play, especially during symbolic activities and pretend play  (see Jarrold, 2003 for a review, Stanley & Konstantareas, 2007).

In addition to this, impairments during play are also present in the third point of the A criterion of the DSM 5, constituting a fundamental element for the ASD diagnosis.>>

  •         Pg 2, Line 64. I understand that poor interaction during play may affect the parent-child relationship (albeit I’m not sure to what degree); however, how is this relationship “restored”? I don’t think that the relationship is “restored”. (This should also be addressed on Pg 3, Line 148).

<<Thanks for the comment. We modified the word in the new manuscript>>

  •         Pg 2, Line 66. I would use a different word than “involving” as it reads awkwardly.

<<Thanks for the comment. We used the word involving to refer to the child’s attempts to involve caregivers during the interplay. The term involving is often used in the literature of children with ASD>>·        

 Pg 2, Lines 69-72. This sentence does not make sense and needs reworded.

<<Thanks for the suggestion. We rephrased the sentence to provide more clarity>>

  •         Pg 2, Lines 74-75. The examples of intrusive behavior are fine, but poorly worded. I would delete “often” or change it to “…calling the child’s name frequently”. “Child’s timing” is unclear.

<<Thanks for the comment. We provided changes as you suggested>>

  •         Pg 2, Line 78. The tense in this sentence is present and should be changed to past tense (i.e., investigated instead of that investigates). Additionally, the rest of the sentence needs rewording. I suggest “A recent study investigated the play behaviors of mothers with children diagnosed with ASD, mothers with children diagnosed with Down syndrome, and mothers of typically-developing children.” Then describe the findings. The current sentence describing the findings should be reworded as well.

<<Thanks for the suggestion. We modified following your comments>>

  •         Pg 2, Line 83. I don’t think “endeavors” fits here.

<<Thanks for the comment. We modified the main text>>

  •         Pg 2, Line 95. I think “has” or “has mostly” should be inserted between “ASD” and “focused”.

<<Thanks for the suggestion. We modified following your comments>>

  •         The tense sometimes shifts from present to past and vice-versa. Please make sure that all sentences maintain the same verb tense throughout the introduction when discussing past research.

<<Thank you for noticing this. We modified the reported results as you suggested>>

  •         Pg 2, begging at Line 148. The review of the literature is focused on parent play behaviors, but in this sentence, it jumps to therapeutic intervention. There needs to be a better explanation of what intervention means. Is the intervention on the parents’ behaviors, the children’s behaviors, or both? What does intervention look like?

<<Thanks for the comment. We added information about parental-based intervention>>

  •         Pg 4, Line 154. Consider changing to “The characteristics of play of children with ASD has received…”. At is currently written the tense is incorrect and the wording is slightly awkward.

<<Thanks for the suggestion >>

  •         Pg 4, Line 174. Add a comma between “mothers” and “consistent”.

<<Thanks for the suggestion >>

Methods:

Minor considerations:

  •         Pg 4, Line 191. Consider beginning the sentence with something other than N=144. You might try “A total of 144…”. Additionally, participants do not “take place” – they participate. The following sentence beginning with “72 children” needs to be reworded.

<<Thanks for the comment. We provided a revised version of these sentences >>

  •         Pg 4, Lines 200-201. “Confirmed” is a better choice than “proved”. Additionally, delete “and judgment”.

<<Thanks for the suggestion. We modified following your comments>>

  •         Pg 5, Lines 203-204. Delete “a golden standard instrument for the diagnosis, i.e.,”.

<<Thanks for the suggestion>>

  •         Pg 5, Line 207. Delete “dedicated”.

<<Thanks for the suggestion>>

  •         The description of the play sessions is much better than before, but could use editing and additional information.

<<Thanks for the comment. We provided all the information about play sessions. If you have specific questions please let us know and we would be more than happy to answer your questions >>

  •         Pg 5. What is the clinical evaluation that was taking place? This should be explained.

<<Thanks for the comment, we provided additional information>>

  •         The description of the Play Code needs additional editing.

<<Thanks for the suggestion, in the description of the play code we provided an explanation of each specific level and an example to make it clear this observational instrument. Further, we also provided references for additional information about the code>>

  •         The Play Code is at times capitalized and at other times it is not. Please use the appropriate form consistently.

<<Thanks for the suggestion. We modified the text in the manuscript>>

  •         Pg 7, Paragraph beginning at 290. This paragraph needs additional editing. Some of the procedures are clear and some are not.

<<Thanks for the suggestion. We provided some changes to provide more clarity throughout the text>>

Results:

Minor considerations:

  •         It is unclear what the W statistic being used is. Is this Kendall’s W? There is also a V statistic that is in the text and table. This needs explanation.

<<According to the R documentation the W is defined as the smaller of W+ (sum of the positive ranks) and W- (sum of the negative ranks), it is referring to the independent test (when we compare mothers' and fathers' behavior during play with their children), and it is equivalent to the Mann-Whitney U statistics. 

The V statistics is is referring to the paired tests when we compare the same children playing with their mothers and fathers and represent the sum of ranks assigned to the differences with positive signs. >>

Discussion:

Minor considerations:

  •         Pg 11, Line 439. I don’t think that the term “intricate” is a good choice. Simply using “difficult” would work. Pg 12, Line 486. Please expound upon the vulnerability to reactivity.

<<Thank you for your comment, we modified the new manuscript following your suggestion>>